# CompLLM: Compression for Long Context Q&A

## Abstract

Large Language Models (LLMs) face significant computational challenges when processing long contexts due to the quadratic complexity of self-attention. While soft context compression methods, which map input text to smaller latent representations, have shown promise, their real-world adoption is limited. Existing techniques typically compress the context as a single unit, which leads to quadratic compression complexity and an inability to reuse computations across queries with overlapping contexts. In this work, we introduce CompLLM, a soft compression technique designed for practical deployment. Instead of processing the context holistically, CompLLM divides it into segments and compresses each one independently. This simple design choice yields three critical properties: efficiency, as the compression step scales linearly with the context length; scalability, enabling models trained on short sequences (e.g., 1k tokens) to generalize to contexts of 100k tokens; and reusability, allowing compressed segments to be cached and reused across different queries. Our experiments show that with a 2x compression rate, at high context lengths CompLLM speeds up Time To First Token (TTFT) by up to 4x and reduces the KV cache size by 50%. Furthermore, CompLLM achieves performance comparable to that obtained with the uncompressed context, and even surpasses it on very long sequences, demonstrating its effectiveness and practical utility.

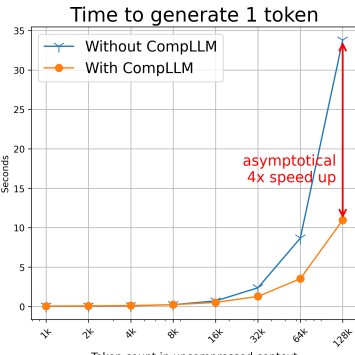

|  | HotpotQA | Musique | NQ | Qampari | Quest |
|---|---|---|---|---|---|
| Gemma3-4B | 0.02 | 0.01 | 0.02 | 0.00 | 0.00 |
| + CompLLM | 0.33 | 0.13 | 0.38 | 0.14 | 0.09 |
| Qwen3-4B | 0.07 | 0.00 | 0.01 | 0.00 | 0.00 |
| + CompLLM | 0.07 | 0.07 | 0.26 | 0.05 | 0.08 |

Table 1: Accuracy across the 5 datasets from LOFT RAG with and without CompLLM. LOFT is a long context benchmark (128k tokens) designed to stress-test the long context capabilities of frontiers LLMs as Gemini 1.5 Pro, GPT-4o, and Claude 3 Opus. With CompLLM we show that we can improve long context capabilities of much smaller open source LLMs.

Figure 1: At high context lengths, CompLLM leads to considerable speedup and improved results, without requiring any modification or tuning of the LLM, by efficiently reducing the number of embeddings fed to the LLM. The plot shows the Time To First Token (TTFT) with CompLLM and without it (*i.e.* with a standard pipeline) as a function of context length. More details in Section 3.3.

## 1 Introduction

Among the many use cases of LLMs, one of the most popular is long context Q&A: given a textual context of arbitrary length, the LLM should answer questions about it. Applications include coding assistants reading large codebases (Team, 2024), web agents reasoning on HTML pages (Zeng et al., 2024), users querying an LLM about a set of documents (Liu et al., 2024a), or RAG systems where LLMs are fed retrieved documents (Lewis et al., 2020a). Due to the quadratic complexity of the transformer (Vaswani et al., 2017), processing long contexts can be unfeasibly expensive: it is therefore important to reduce computational complexity, especially as contexts grows longer.

To reduce computational complexity, prior works in literature focus on compressing long contexts into smaller representations that can be passed to the LLM, while aiming to obtain similar outputs. These works can be split into two categories: the first category aims to compress the context into shorter text (also called hard compression), often by pruning low-entropy or non-informative tokens or sentences (Jiang et al., 2023; Xu et al., 2024); while the second category compresses the prompt into a high-dimensional latent space (also called soft compression), either in the form of embeddings (Li & Liang, 2021; Wang et al., 2024) or in the form of KV cache (Petrov et al., 2025a; Mu et al., 2023). While hard compression allows for higher interpretability, given that the compressed representation is human readable, soft compression produces continuous representations, allowing end-to-end training and providing higher flexibility. Furthermore, soft compression representations do not need to belong to the domain of natural text, enabling higher compression rates than token pruning (Corallo & Papotti, 2024) and leads to higher quality outputs, often on par with the non-compressed pipeline (Ge et al., 2024).

These soft compression methods have achieved increasingly better performances, enabling increasingly higher compression rates with little accuracy drop. But despite these advances, adoption of soft compression methods in real-world applications is still scarce. In this paper we present CompLLM, a new soft compression technique which, instead of aiming for high compression rates, focuses on satisfying a set of properties that are necessary for widespread real-world adoption. CompLLM aims to compress text into fewer tokens that can be consumed by the original unmodified LLM.

Existing soft compression methods compress the context as a whole, meaning that every input token affects the entire compressed representation; on the other hand, we propose to split the context into segment (*i.e.* short sentences) which are compressed independently. This simple design choice naturally leads to three important properties of CompLLM:

1. **efficiency**: while in existing methods a token attends to every previous one in the attention layers used for compression (due to the context being compressed as a whole), which leads to quadratic complexity, in CompLLM each token only attends to previous tokens within its segment. This makes the computational complexity of the compression step linear w.r.t. the number of segments, and hence the length of the context.
2. **scalability**: CompLLM can be trained on shorter contexts than those used at test time, given that CompLLM effectively only sees small chunks (segments) of context at a time. In practice, we show that despite our model being trained on sequences no longer than 2K tokens, it can compress contexts of hundreds of thousands of tokens while retaining (or even improving) results derived from the uncompressed prompt.
3. **reusability**: CompLLM's compressed representations can be reused across queries. Imagine asking an LLM to compare documents A and B, and in a second query asking it to compare documents A and C: because the compressed representation of A is independent from that of B, such representation can be reused for the second query. This can be useful for systems where contexts are often reused, like (i) applications where the context comes from a predetermined set of documents, as in RAG systems, and (ii) applications where the majority of the context does not change, like coding assistants for large codebases.

To evaluate the efficacy of CompLLM, we provide evaluations on multiple LLMs, datasets, and context lenghts. Our results highlight a number of interesting empirical qualities of our CompLLM, which uses a compression rate of 2:

1. CompLLM speeds up Time To First Token (TTFT) by up to 4x for long contexts;
2. CompLLM reduces the size of the KV cache by 2x;
3. CompLLM achieves results competitive to a standard LLM pipeline (*i.e.* without compression) for short context lengths, while leading to better results at long context lengths. We hypothesize that this happens because having fewer tokens reduces attention dilution.

## 2 RELATED WORK

Among the multiple works that approached the task of reducing the inference cost for LLMs through compression, it is possible to identify two categories: *hard compression*, *i.e.* methods that compress the prompt into shorter prompts in natural language, and *soft compression*, which compresses prompts into various forms of latent representations.

## 2.1 HARD COMPRESSION

These methods aim to synthesize prompts into shorter ones in natural language, either through means of token pruning (Jiang et al., 2023; Pan et al., 2024; Chung et al., 2024), sentence pruning (Xu et al., 2024), or paraphrasing the prompt (or context) (Ali et al., 2024; Yang et al., 2023). While some of these methods are question-agnostic, a large number of question-aware methods have been developed, either for sentence or document pruning (Hwang et al., 2025; Liskavets et al., 2025; Zhao et al., 2025b; Fei et al., 2025) or for token pruning (Zhao et al., 2025a; Tang et al., 2025; Jiang et al., 2024b). These methods have the advantage of being interpretable, and usable with closed LLMs through API, as it is possible to compress the prompt locally and send only the synthesized prompt to the LLM. However, these usually result in lower compression rates, and incur into higher accuracy drops, compared to their counterparts that compress prompts in latent space (Liu et al., 2024b; Chen et al., 2025).

## 2.2 SOFT COMPRESSION

Soft compression is achieved in two different ways: (A) by compressing text into latent embeddings ($N_1$ $D$-dimensional embeddings, where $N_1$ is the sequence length of the compressed representation) (Ge et al., 2024), and (B) methods that compress the input into a key-value (KV) cache (Li et al., 2025), which has dimension $N_2 \times L \times D \times 2$, where $N_2$ is the sequence length of the generated KV cache, $L$ is the number of layers in the LLM, $D$ is the dimension of the latent embeddings, and 2 is due to each token requiring one embedding for the key and one for the value.

Compressing into KV cache can generally lead to shorter sequence lengths: this led to the development of multiple works along these lines (Chari et al., 2025; Kim et al., 2024; Liu et al., 2024b; Petrov et al., 2025b), with Li et al. (2025) pushing the compression to the limit with hundreds of tokens compressed into a KV-cache of sequence length 1, and Corallo & Papotti (2024) aim to get higher accuracy by building question-aware KV-cache representations. It must be noted that, in the KV cache, each key-value embedding depends on all its preceding tokens: this makes KV cache a holistic latent representation by design, *i.e.* the representation of different sentences in the context can't be independent from each other, leading to non-reusability, and non-linear scalability (time complexity of $O(N^2)$). On the other hand methods that reduce computational complexity by employing sparse attention masks, which could be static Beltagy et al. (2020); Zaheer et al. (2020) or dynamically determined based on the input Liu et al. (2022); Jiang et al. (2024a), do not actually compress the KV cache, and thus do not provide a reduction in the KV cache memory footprint during inference.

Among the most similar works to ours, several papers compress prompts into latent embeddings: (Li & Liang, 2021) spearheaded the task by generating a set of latent embeddings for each of a few tasks, like summarization or translation; similarly, (Mu et al., 2023) proposed to directly compress prompts into latent embeddings. Chevalier et al. (2023) aimed at achieving longer context windows by iteratively compressing and accumulating the context into summary vectors. (Ge et al., 2024) proposed a model to compress any context into a fixed sequence of latent concept embeddings, into what they call *memory slots*, without fine-tuning the LLM used for generation: this has inspired a number of subsequent papers, like Wang et al. (2024), which uses a perceiver-like architecture Jaegle et al. (2021) to compress the context, Cao et al. (2024) which creates a query-dependent compression, Huang et al. (2024) which recursively compresses context of increasingly larger lengths, Cheng et al. (2024) which pushes compression to the extreme into a single latent embedding. Despite the large number of work in this area, all of these compress the context as a block, which despite allowing higher accuracy and compression rates, does not allow the properties *efficiency*, *scalability*, and *reusability* (see Section 1).

# 3 METHOD

## 3.1 COMPLLM

In this work we propose CompLLM (**Comp**ression for **LLM**s), which reduces computational complexity by reducing the number of embeddings fed to the LLM. In a standard setting, an LLM can be fed one of roughly 200k Token Embeddings (**TEs**), *i.e.* the vectors contained in the embeddings

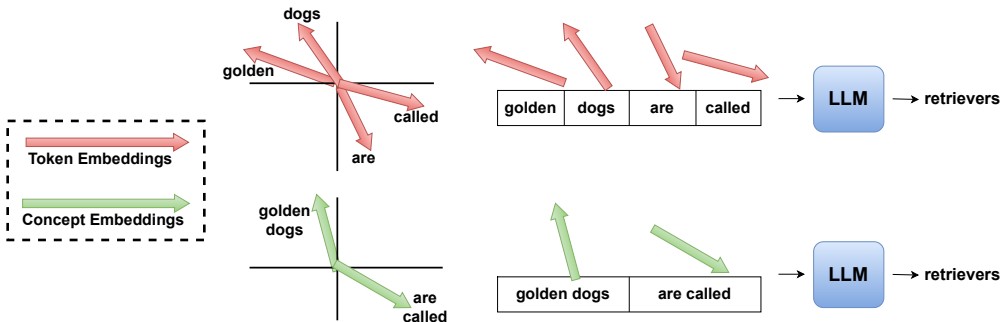

Figure 2: Conceptualization of *Token Embeddings* (TEs) (Top) and *Concept Embeddings* (CEs) (Bottom), and how they can both lead to the same output, using the sentence "*golden dogs are called*" as an example. TEs are contained in the LLM's embeddings table and limited to roughly 200k (*e.g.* 262k for Gemma3 models and 151k for Qwen3 models). CEs lie in the same features space as TEs, but are not limited in number, and can be fed directly to the LLM without tuning it. The sentence *golden dogs are called* can be represented with 4 TEs, or in a more compact way using 2 CEs, while leading to the same output. A CompLLM's objective is to extract CEs given TEs, in order to reduce the computational burden on the LLM.

table: for example, Gemma3 (Team et al., 2025) has 262k TEs, while Qwen3 (Yang et al., 2025) has 151k tokens. We instead rely on the existence of other embeddings, which we call Concept Embeddings (**CEs**, conceptualized in Figure 2), which exist in the same latent space of TEs and can be directly fed to the LLM, despite being completely unseen at training time. CEs allow to encode a similar amount of information as TEs, leading to similar outputs, while reducing the sequence length, which reduces latency and memory usage of the LLM's forward pass.

While TEs are chosen from a finite vocabulary of embeddings in the embeddings tables, CEs need to be efficiently estimated from the input text using a specialized model. To this end, given a text of length $N$ tokens, we split it into *segments* of maximum length $S$ tokens, obtaining $\frac{N}{S}$ segments. Each segment is passed to the CompLLM independently, so that the attention operation is quadratic within each segment, but scales linearly over the whole context. The complexity within each segment is $O(S^2)$, therefore for $\frac{N}{S}$ segments overall complexity is $O(\frac{N}{S}S^2) = O(NS)$. In practice, we set $S = 20$, meaning that for a compression rate of $C = 2$ we compress each *segment* of 20 TEs into 10 CEs. CompLLM's architecture therefore needs to be able to take as an input S (or fewer) embeddings and output $\frac{S}{C}$ embeddings: note that multiple architectures can satisfy this basic constraint (like encoder-only LLMs, decoder-only LLMs, MLPs, etc. ).

As an architecture for our CompLLM, we take inspiration from Ge et al. (2024), by attaching a LoRA (Hu et al., 2022) to the same LLM used for generation; on top of it, we append a single linear layer. Specifically, when feeding this CompLLM with a sequence of length $S$, we append $\frac{S}{C}$ embeddings corresponding to *EOS* tokens, whose corresponding outputs are used as the $\frac{S}{C}$ CEs, as shown in the bottom left of Figure 3. This simple architecture has the advantage of reusing the parameters of the LLM, which are left untouched by the LoRA, hence reducing memory usage needed to store weights; this also allows to use the LLM in the standard fashion (*i.e.* without CompLLM) when needed. Finally, we emphasize that benchmarking different architectures as a choice for compressor is outside of the scope of this manuscript, whereas our goal is instead to showcase that CompLLM is a feasible and useful alternative to the standard LLM pipeline.

## 3.2 TRAINING

Among the possible applications of CompLLMs, the most helpful one is long-context question answering, which is the focus of our training and evaluation experiments. To this end, we design our pipeline to reflect the real-world scenario, where compression is used on the long context (and can optionally be computed offline), whereas the question (much shorter than the context, and provided online) is not compressed. Building on these considerations, we now describe how we train CompLLM to handle long-context question answering effectively.

Consider an instruction-tuned LLM $p_{\text{LLM}}(y \mid c, x)$ where $c$ represents context, $x$ the instruction and $y$ the generated response. Our goal, conceptualized in Figure 3, is to fine-tune a compressor $p_{\text{CompLLM}}$ (CompLLM) that maps $c$ to a compressed context $\widehat{c} = \text{CompLLM}(c)$, where $c$ is made

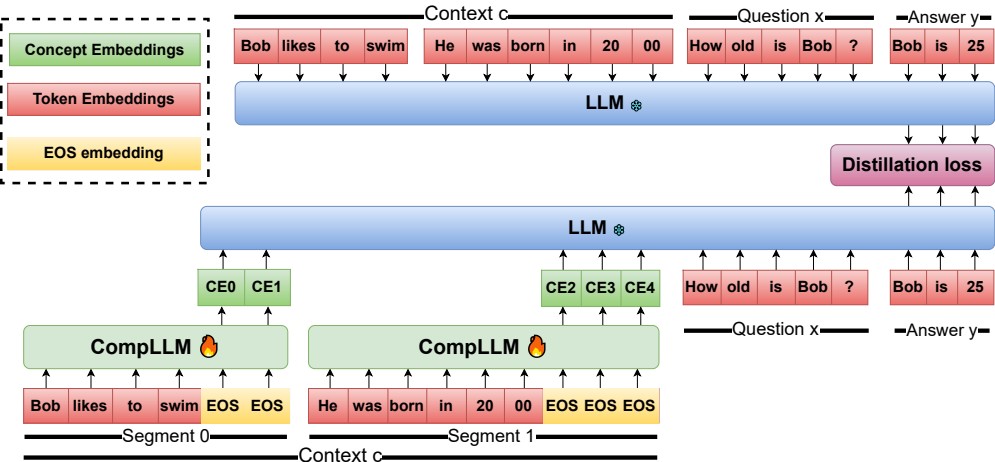

Figure 3: Training protocol of CompLLM for context-based Q&A. The CompLLM can extract multiple CEs in a single forward pass, and can take as input any number of segments with any number of TEs and output any number of CEs (*i.e.* the number of CEs is proportional to the number of TEs). The loss is computed only on the activations corresponding to the answer's embeddings, and the outputs corresponding to the other embeddings are ignored. The answer *y* can be computed online (during training) or offline; here we show it as if it was pre-computed for simplicity.

of TEs and $\widehat{c}$ is made of CEs. The base LLM is queried with either $c$ or $\widehat{c}$. Instead of matching output distributions, we distill by matching *hidden activations* on the answer segment, which provides a denser and richer signal than output distributions.

Let $A$ denote the indices of the answer tokens (the last $|A|$ tokens), and let $H_A^{(\ell)} \in \mathbb{R}^{|A| \times d}$ be the teacher hidden states at layer $\ell \in \{1, \ldots, L\}$ restricted to $A$; $\widetilde{H}_A^{(\ell)}$ are the corresponding student states obtained when conditioning on $\widehat{c}$. We minimize a Smooth-$L_1$ loss per layer, normalized by the scale of each layer's teacher activation:

$$\mathcal{L}_{\text{layer}}^{(\ell)}(c, x) = \frac{1}{\sigma^{(\ell)}(c, x)} \frac{1}{|A| \, d} \sum_{t \in A} \sum_{j=1}^{d} \text{SmoothL1}_\beta\left(\widetilde{H}_{t,j}^{(\ell)}, H_{t,j}^{(\ell)}\right), \tag{1}$$

$$\sigma^{(\ell)}(c, x) = \text{Std}\left(H_A^{(\ell)}\right), \quad \text{SmoothL1}_\beta(u, v) = \begin{cases} \frac{1}{2}(u-v)^2/\beta, & |u-v| < \beta, \\ |u-v| - \frac{\beta}{2}, & \text{otherwise,} \end{cases} \tag{2}$$

with $\beta=1$ in our experiments (*i.e.* PyTorch's default Paszke et al. (2019)). The normalization allows to compensate for large cross-layer activation-norm variability, following (Shen et al., 2025). The training objective is the expectation over context–instruction pairs:

$$\mathcal{L}_{\text{comp}}(p_{\text{CompLLM}}, \mathcal{CX}) = \mathbb{E}_{(c,x) \sim \mathcal{CX}}\left[\sum_{\ell=1}^{L} \mathcal{L}_{\text{layer}}^{(\ell)}(c, x)\right]. \tag{3}$$

This loss aligns the internal representations for the *answer tokens* produced with $\widehat{c}$ to those produced $c$, encouraging the compressed context to preserve information essential for generation. No ground-truth labels are required; $y$ (and thus $A$) is obtained from the LLM during training.

Both at training and test time, we use CompLLM to compress the contexts, but do not compress the question. This matches the use case that CompLLM would face in a real-world scenario: compressing the context is useful because (A) contexts are long and (B) for many applications context can be compressed offline; on the other hand questions are generally short and provided online. This means that the LLM receives both CEs (context) and TEs (question), as shown in Figure 3.

## 3.3 COMPUTATIONAL COMPLEXITY ANALYSIS

The computational cost of LLM inference can be divided into two components. The first is the *KV cache prefill* cost, incurred when computing the first forward pass over the input tokens, and is virtually equivalent to the Time To First Token (TTFT). This cost scales quadratically ($O(N^2)$)

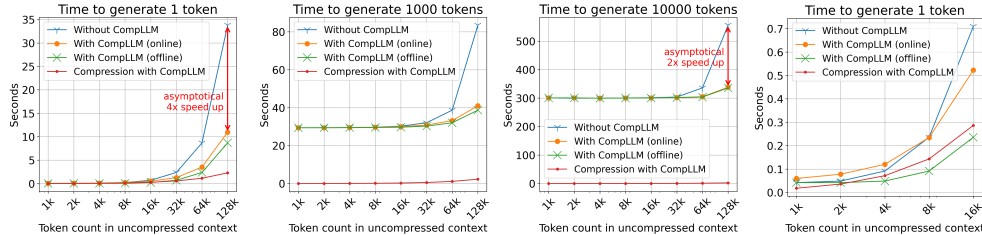

Figure 4: Inference speed with and without CompLLM, for contexts of different lengths (x axis), for different number of generated tokens (1, 1k, and 10k in the first 3 plots respectively), for a CompLLM with compression rate $C = 2$. The rightmost plot is a zoomed-in version of the first plot, to show the crossover point, *i.e.* the point over which compression leads to faster Time To First Token (TTFT) *even when performed online*. Generation with online CompLLM compression (orange line) equals generation with offline compression (green) plus compression time (red). We used Gemma3-4B on a B200 GPU with BFloat16 and PyTorch compile. **The leftmost plot** shows TTFT: the latency ratio between *with* and *without* CompLLM asymptotically approaches 4x (*i.e.* $C^2$), and the compression time asymptotically becomes negligible (as it scales linearly, not quadratically like the other 3 curves) **The third plot** shows the time taken to generate 10k tokens, where *next token prediction* time overcomes *KV cache prefill* time, asymptotically bringing the ratio to 2x ($C$).

with the prompt length $N$ for standard attention-based LLMs (Vaswani et al., 2017). The second component is the *next token prediction* cost, which, considering the KV cache to be already prefilled, corresponds to producing new tokens in an autoregressive manner. Each generated token attends to all previously processed tokens, leading to a per-token complexity of $O(N)$, which to generate $T$ tokens becomes $O(NT)$ [1]. Consequently, total inference complexity depends on both the initial context length and the number of generated tokens.

For CompLLMs, the *KV cache prefill* cost scales down quadratically with $C$: from $O(N^2)$ it drops to $O(\frac{N^2}{C^2})$. The *next token prediction* cost instead scales down linearly by $C$, from $O(NT)$ to $O(\frac{N}{C}T)$, because the number of tokens that each new token should attend to is divided by $C$. Furthermore, CompLLMs also incur in one additional cost, the *compression time*: however, due to its linear complexity of $O(NS)$ (see Section 3.1), *compression time* becomes negligible for large $N$, given that *KV cache prefill* has quadratic complexity and $O(NS) + O(\frac{N^2}{C^2}) = O(\frac{N^2}{C^2})$. This is empirically shown in Figure 4. Moreover, compression can be computed offline in many real-world cases (*e.g.* in a RAG pipeline the documents are likely to be available beforehand). We ground these concepts into empirical results in Figure 4, where the plot shows the latency of *next token prediction* at different context lengths, with different number of generated tokens, with and without CompLLM.

From a computational complexity perspective, there are 3 types of situations:

1. for large $N$ and small $T$ the *KV cache prefill* cost is the bottleneck, meaning CompLLM can decrease inference time by $C^2$;
2. for large $N$, as $T$ grows larger, the computational gains from CompLLM approach $C$, as the *next token prediction* cost begins to overcome the *KV cache prefill* cost.
3. for very small $N$, CompLLM will actually slow down generation, as for small $N$ the compression time is non-negligible. Note however that in these cases CompLLM can be seamlessly unplugged from the pipeline, as the LLM's weights are untouched by CompLLM.

## 4 EXPERIMENTS

### 4.1 IMPLEMENTATION DETAILS

We implement CompLLM with two recent instruction-tuned LLMs, Gemma3-4B (Team et al., 2025) and Qwen3-4B (Yang et al., 2025) as base LLM. CompLLM's LoRA uses a rank of 32, following (Ge et al., 2024). We use Adam (Kingma & Ba, 2015) with learning rate 0.0001, and train with batch size 4 until convergence. Unless otherwise specified, we use a compression rate of $C = 2$. We split

---

[1] More precisely, the per-token complexity for *next token prediction* is $O(N+T)$, hence $O(T(N+T))$ for $T$ tokens, as each new token attends not only the ones from the prompt, but also the newly-generated ones. Given that in long-context Q&A we have that $N >> T$, we simplify notation by using $O(N)$ instead of $O(N + T)$

| Dataset | Type | Split | # samples | Context Len. | | | Question Len. | | | Answer Len. | | |
|---------|------|-------|-----------|-----|-----|-----|-----|-----|-----|-----|-----|-----|
| | | | | avg | min | max | avg | min | max | avg | min | max |
| NarrativeQA | OE | train | 32.7k | 743.4 | 242 | 1395 | 11.0 | 3 | 41 | 6.3 | 1 | 36 |
| | | valid. | 3.5k | 719.3 | 248 | 1293 | 10.8 | 4 | 31 | 6.1 | 1 | 50 |
| | | test | 10.6k | 734.2 | 249 | 1349 | 10.9 | 4 | 34 | 6.2 | 1 | 42 |
| SQuAD | OE | test | 10.6k | 167.9 | 30 | 788 | 12.8 | 4 | 40 | 4.8 | 1 | 50 |
| RACE | MC | train | 87.9k | 341.7 | 4 | 1436 | 98.0 | 66 | 414 | 1 | 1 | 1 |
| | | valid. | 4.9k | 337.5 | 60 | 1063 | 97.7 | 68 | 161 | 1 | 1 | 1 |
| | | test | 4.9k | 339.4 | 23 | 1048 | 98.1 | 70 | 188 | 1 | 1 | 1 |
| QuAIL | MC | test | 0.6k | 420.2 | 376 | 501 | 86.5 | 71 | 116 | 1 | 1 | 1 |

Table 2: Statistics for each dataset, including context, question, and answer lengths, in number of tokens. *OE* means Open-Ended questions, while *MC* stands for Multiple Choice questions.

the text into segments using the NLTK Punkt tokenizer (Kiss & Strunk, 2006), a classical algorithm, and split again any long sentence to ensure they are shorter than $S = 20$ tokens. We found no benefit by using a learnable embedding instead of EOS token for CEs; we also found that interleaving CEs within the segments (following Petrov et al. (2025a)) leads to a non-negligible drop in accuracy.

## 4.2 DATASETS

To validate CompLLM's capabilities, we conduct experiments on long-context Q&A datasets. Specifically, we use two open-ended Q&A datasets, namely NarrativeQA (s Kočiský et al., 2018) and SQuAD (Rajpurkar et al., 2016), and two multiple choice Q&A datasets, namely RACE (Lai et al., 2017) and QuAIL (Rogers et al., 2020). We use one dataset of each kind (NarrativeQA and RACE) for training, and test on all four datasets, to assess the generalization capabilities of CompLLM. For open-ended Q&A, we compute evaluation with the LLM-as-a-judge approach; for multiple choice Q&A, we evaluate with regex matching, and prompt the LLM to end its chain-of-thought output with *the answer is (X)* [2], where X is one of A, B, C or D. Statistics for each dataset are shown in Table 2, and examples of the datasets are shown in Appendix A.1. Based on the numbers in Table 2, we can infer that the contexts in the training set contain altogether $32.7k \times 743.4 + 87.9k \times 341.7 = 54M$ tokens. The number of tokens of the generated answers (used for distillation) are on average 149 for Gemma3-4B and 273 for Qwen3-4B, hence 18M and 33M respectively.

Furthermore, we evaluate on the suite of RAG datasets from LOFT (Lee et al., 2024), where, similar to the open-ended Q&A datasets described above, the goal is to answer an open-ended question given a long context. LOFT defines each dataset as a collection of 100 questions about a context with length of 128k tokens, using the datasets of HotpotQA (Yang et al., 2018), Musique (Trivedi et al., 2022), NQ (Kwiatkowski et al., 2019; Thakur et al., 2021), Qampari (Amouyal et al., 2023), and Quest (Malaviya et al., 2023).

## 4.3 MAIN RESULTS

We compare CompLLM to the baseline LLM, *i.e.* with no compression, to show that CompLLM is able to produce comparable results with the advantage of reducing computational needs. We showcase results on context-based open ended Q&A, and context-based multiple choice Q&A.

We compute results across the four datasets' (NarrativeQA, SQuAD, RACE, QuAIL) test sets, after training CompLLM on two of their train sets (NarrativeQA and RACE). To understand how well CompLLM works with contexts of different length, we compute results both in a standard fashion (evaluating independently each context/question/answer triplet) and simulating longer contexts through concatenation, following LOFT's paradigm. Specifically, we concatenate $N_{ctx}$ contexts together, and evaluate the questions-answers independently over this longer context. Starting with $N_{ctx} = 1$ (single context), we gradually increase it as long as the number of embeddings (either TEs or CEs) fed to the LLM is lower than 128k. Note that CompLLM's compression allows it to process longer raw contexts for any given limit in number of embeddings.

Results are reported in Figure 5: it can be seen how using CompLLM leads to comparable or slightly lower results at lower context lengths, while achieving on average better results at higher context

---

[2]As showcased in Appendix A.1, we use the following prompt for multiple choice Q&A: *The following is a multiple choice question (with answers), about the above text. Think step by step and then make sure to end your answer with "the answer is (X)" where X is the correct letter choice.*

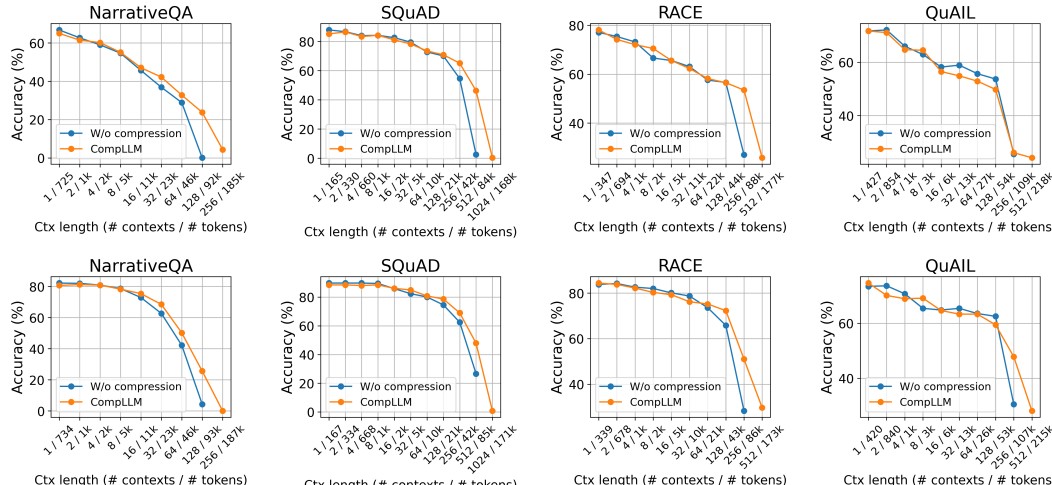

Figure 5: Results with and without compression across multiple context lengths for four datasets, with Gemma-3-4B (top row) and Qwen3-4B (bottom row). X axis indicates the context length, showing both the number of concatenated contexts/documents and number of tokens (before compression). To obtain longer contexts, multiple contexts are concatenated, and the correlated questions (one per context) are asked independently from each other: this means that the distribution of the relevant contexts is uniform within the concatenated context - there is exactly one question about the first document, one question about the second document et cetera.

lengths. It should be noted that not only does CompLLM achieve better results at high context lengths (*i.e.* over 50k tokens), it also does so while asymptotically reducing latency for *cache prefill* by 4x (see Section 3.3), reducing latency for *next token prediction* by 2x and reducing the KV cache by 2x. A thorough analysis on the computational complexity is available in Section 3.3.

## 4.4 RESULTS ON LOFT

The Long-Context Frontiers benchmark (LOFT) (Lee et al., 2024) is a recent benchmarks designed to assess LLMs' performance on long context tasks across a variety of datasets, described in Section 4.2. Similarly to our experiments above, LOFT uses a set of context-based questions and answers, and concatenates the contexts; furthermore, LOFT adds distractors contexts to reach a length of 128k tokens. Results with and without CompLLM are reported in Table 1. We emphasize that this benchmark was designed to compare the long-context capabilities of the frontiers models of Gemini 1.5 Pro (Team, 2024), GPT-4o (OpenAI et al., 2024), and Claude 3 Opus (Anthropic, 2024), hence these are tasks that prove very challenging for smaller models: nonetheless, CompLLM is able to always match or improve the results reached by the baseline, while providing a significant performance enhancement, as more thoroughly described in Section 3.3

## 4.5 COMPARISON WITH EXISTING TECHNIQUES

The vast majority of compression methods compress the context as a whole, making the compressed representations intrinsically holistic and non reusable, and leading to quadratic complexity, (see Table 3 and Section 2), making them unsuitable for long context compression and not directly comparable with CompLLM. As an outlier, LLMLingua-2 (Pan et al., 2024) uses a BERT-like encoder (Devlin et al., 2019), which compresses sentences independently from each other: this makes LLMLingua-2 scale linearly with the context length, making it a fit choice for long context compression, and allowing reusability of compressed representations. In a similar fashion, LongLLMLingua (Jiang et al., 2024b) also compresses the context in linear complexity, although its architecture takes as input the question together with the context: this makes the compressed representation question-dependent, hence non-reusable. Somewhat differently, AutoCompressor (Chevalier et al., 2023) instead compresses the context iteratively: after splitting the context in segments, these are compressed one-by-one with the output of the previous segment's compression being fed as input to help compress the next segment. This means that the first token in the context influences the en-

| Method | Dynamic compression size | Preserves LLM weights | Reusability | Soft Compression | # sequential forward passes | Compression Complexity |
|---|---|---|---|---|---|---|
| Prefix-Tuning Li & Liang (2021) | ✗ | ✓ | ✗ | ✓ | 1 | $\mathcal{O}(n^2)$ |
| AutoCompressor Chevalier et al. (2023) | ✓ | ✗ | ✗ | ✓ | $\mathcal{O}(n)$ | $\mathcal{O}(n)$ |
| Gist Tokens Mu et al. (2023) | ✗ | ✗ | ✗ | ✓ | 2 | $\mathcal{O}(n^2)$ |
| ICAE Ge et al. (2024) | ✗ | ✓ | ✗ | ✓ | 2 | $\mathcal{O}(n^2)$ |
| LLMLingua-2 Pan et al. (2024) | ✓ | ✓ | ✓ | ✗ | 2 | $\mathcal{O}(n)$ |
| LongLLMLingua Jiang et al. (2024b) | ✓ | ✓ | ✗ | ✗ | 2 | $\mathcal{O}(n)$ |
| Gist of Gisting Petrov et al. (2025b) | ✓ | ✗ | ✗ | ✓ | 2 | $\mathcal{O}(n^2)$ |
| **Ours** | ✓ | ✓ | ✓ | ✓ | 2 | $\mathcal{O}(n)$ |

Table 3: Overview of various compression methods. CompLLM is the only soft compression method that preserves LLM weights and has linear compression complexity.

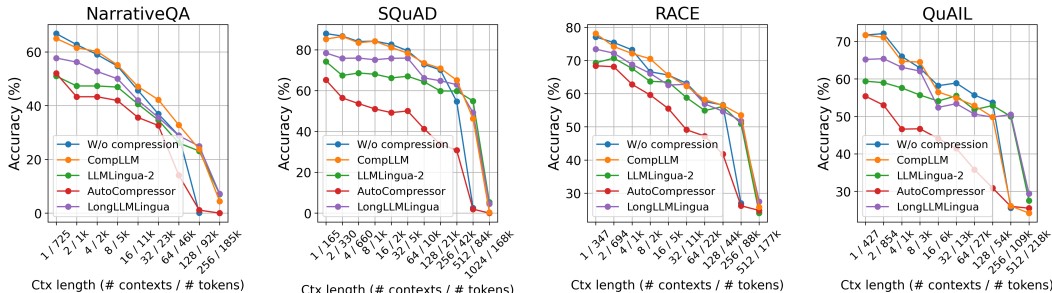

Figure 6: Results with Gemma3-4B with no compression, CompLLM, LLMLingua-2, LongLLM-Lingua and AutoCompressor, at different context lengths.

tire compressed representation, making each segment's representation non-reusable. Given these observations, we compute experiments to compare results between Gemma3-4B without any compression, with CompLLM, with LLMLingua-2, as well as LongLLMLingua and AutoCompressor for completeness. The compression rate is set to 2 for all methods. This leads the asymptotic latency of the pipelines with CompLLM, LLMLingua-2 and LongLLMLingua to be equivalent. Results are reported in Figure 6. We can see that for short to medium context lengths, CompLLM is the only method that is on par with the uncompressed LLM; LongLLMLingua constantly outperforms LLMLingua-2, although using the advantage of taking as input the question besides the context; AutoCompressor performs better on datasets closer to its training distribution, while underperforming on RACE and QuAIL).

## 4.6 ABLATIONS

**Compression rate $C$** is the crucial parameter that directly influences the asymptotical memory footprint and latency of CompLLM: in this paragraph we investigate how it also influences results. Due to lack of space, we report results in Figure 7 in the Appendix; results show that CompLLM achieves graceful degradation of accuracy as the compression rates increases. To cite some examples, $C=4$ leads to a 4% drop in results against the uncompressed baseline with using a single context, and $C=8$ leads to a drop of 15% when concatenating 64 documents as context (roughly 26k tokens) while providing an asymptotical speed up of 64 times.

**Segment length $S$.** In this section we compute results varying the maximum segment length $S$. On one hand, longer segments make compression easier, as it becomes more likely that any given segment contains easy-to-compress redundancy. On the other hand longer segments also increase the train/test distribution gap: with longer segments it becomes ever less likely to have seen the same segment during training, whereas if all segments are very short (*e.g.* 4 words) having encountered any given segment at training time becomes more expected. Furthermore, using longer segments also increases computational complexity: at its extreme, a segment can be as long as the whole context, leading to $\mathcal{O}(n^2)$ asymptotical compression complexity, as in many previous work (see Table 3). Due to lack of space we report results in the Appendix in Figure 8, showing that in practice CompLLM is robust to variations of $S$, with no noticeable improvement or degradation between values of $S$ of 10, 20, and 40.

**Does CompLLM reduce attention dilution?** Attention dilution is the phenomenon for which, as the context length becomes longer, LLMs find it harder to attend to the tokens that are most relevant to the question, especially as relevant tokens might be tens of thousands of positions away from the token that is being predicted. CompLLM reduces the number of keys and values vectors, making it easier to find and attend to the relevant tokens, hence reducing attention dilution. We show empirical results supporting this theory in Figure 9 in the Appendix, where we report the attention scores at different context lengths with and without CompLLM.

### 4.7 LIMITATIONS

CompLLM ensures that the CEs encode the semantic content of a text, not its structure: hence CompLLM by design would not work well for tasks like "count how many times the letter R appears in the text" or "find the typos in this document", as words like "with" and "wiht" (note the typo) are likely encoded with a similar CE. We note however that (1) these tasks are unfeasible for virtually any LLM-based compression method, (2) these tasks can still be tackled within our pipeline, because CompLLM can be seamlessly unplugged (the LLM is frozen) and (3) these non-semantic use-cases represent a small minority of the total use cases of LLMs in the real world.

Another limitation of CompLLM, as well as every other soft compression methods, lies in its inability to handle out-of-distribution data: a CompLLM trained on only on English language would not be able to compress Chinese characters or coding. We emphasize however that CompLLM can simply overcome this issue by not using the compression in out-of-distribution cases (*e.g.* tokens unseen during training), while many existing soft compression methods can't skip the compression stage as they do not preserve the original LLM's weights (Chevalier et al., 2023; Petrov et al., 2025b).

### 4.8 FUTURE WORK

The inception of CompLLM opens up a very wide range of possible future works, outside the scope of this paper, which can lead to increasing the compression rate, achieving better results, speeding up inference, and broadening the scope of CompLLM. Among these, the most noteworthy are:

1. experimenting with dynamic compression rates, *i.e.* the compression rate should depend on the input: complex sentences would benefit from lower compression rates, whereas simple repetitive sentences can be compressed into fewer embeddings.
2. understanding how far can the compression rate be pushed, and how this depends on other factors such as the model size or its features dimension. For example, it is possible that larger models can accommodate higher compression rates, as their embeddings lie in a higher dimensional space.
3. testing different architectures for CompLLM, such as encoder-only models or fully-tuned LLMs (instead of LoRA applied to an LLM).
4. leveraging plain text (instead of context-question pairs) to train CompLLM, to unlock much larger training sets.

## 5 CONCLUSION

In this paper we introduced CompLLM, a technique that allows LLMs to generate up to 4 times faster answers, while simultaneously reducing the memory footprint of the KV cache by 2x and producing outputs of similar or better quality, specifically designed for long-context Q&A. We showed that CompLLM works by extracting new embeddings which can be directly fed to the LLM without fine-tuning it. CompLLM compresses segments of text individually, not only leading to linear computational complexity with the length of the text, but also allowing it to compress documents offline, as their compressed representations can be utilized regardless of other documents in the context and regardless of the question. Notably, CompLLM works across a variety of LLMs and datasets, and is completely orthogonal to many common inference-time techniques, like chain of thought (Wei et al., 2022), RAG (Lewis et al., 2020b), beam search (Freitag & Al-Onaizan, 2017), paged attention (Kwon et al., 2023), et cetera. Finally, we highlight a number of possible future directions, and envision for CompLLM to be directly integrated into major LLMs, which would lead to lower latency, FLOPs, and energy consumption.

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

# A  APPENDIX

In this appendix we show further ablations and qualitative results. Specifically, we test different values of compression rate $C$ in Figure 7, with $C = 2$, $C = 4$, and $C = 8$, finding that as expected higher compression rates lead to a degradation of results, although without leading to a steep drop when using higher compression rates.

In Figure 8 we provide experiments when changing the maximum segments length $S$, testing values of $10, 20, 40$, while keeping $C = 2$, finding that small changes in $S$ do not have a strong impace on the results. We note however that increasing $S$ requires more memory to train: while training Gemma3-4B with $S = 20$ takes only 57GB of GPU memory, with $S = 80$ it does not fit into 80GB.

In Figure 9 we report the average attention scores between the generated tokens and the tokens in the correct (or gold) document in the context. It should be noted that for most generated token the majority of attention is paid to previously generated tokens and to the tokens in the question. The plot shows that CompLLM does help to reduce attention dilution, *i.e.* the phenomenon due to an increasing difficulty for the transformer to "pay attention" to relevant tokens due to a large number of tokens in the context.

In the next section we provide a summary of qualitative results with and without CompLLM, on each of the four main test sets we use in the paper.

## A.1  EXAMPLES OF GENERATED ANSWERS

In this section we show qualitative examples of generation with and without the CompLLM (Appendix A.1), for the open-ended Q&A datasets of NarrativeQA Appendix A.1.1 and SQuAD Appendix A.1.2 and for the multiple-choice Q&A datasets of RACE Appendix A.1.3 and QuAIL Appendix A.1.4. For multiple-choice datasets, we prepend to the question the following text, as shown in the examples below: *The following is a multiple choice question (with answers), about the above text. Think step by step and then make sure to end your answer with "the answer is (X)" where X is the correct letter choice.*. For open-ended datasets, no extra prompt is used other than the context and question (*i.e.* we do not prompt the models to reason step by step or any other type of elaborate prompts). All the following examples belong to the test sets of the respective datasets.

### A.1.1  EXAMPLE FROM NARRATIVEQA

**Context:** Natalie Cook (Cameron Diaz), Dylan Sanders (Drew Barrymore) and Alex Munday (Lucy Liu) are the "Angels", three intelligent, talented, tough, attractive women who work as private investigators together for an unseen millionaire named Charlie (voiced by John Forsythe). Charlie uses a speaker in his offices to communicate with the Angels, and his assistant Bosley (Bill Murray) works with them directly when needed. Charlie assigns the Angels to find Eric Knox (Sam Rockwell), a software genius who created a revolutionary voice-recognition system and heads his own company, Knox Enterprises. Knox is believed to have been kidnapped by Roger Corwin (Tim Curry), who runs a communications-satellite company called Redstar. The Angels infiltrate a party

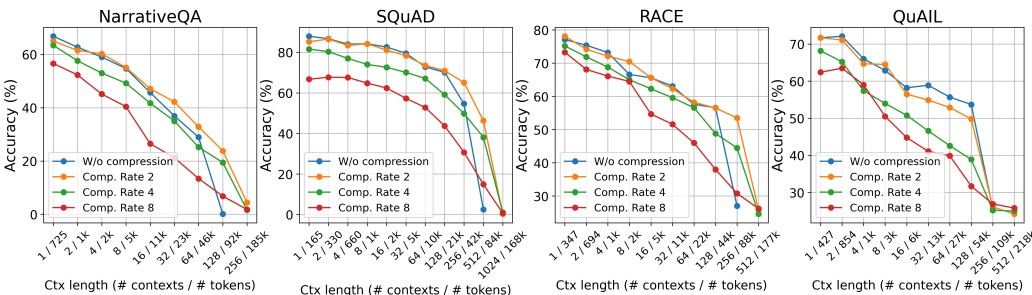

Figure 7: Ablation on compression rate, with no compression and CompLLM with $C = 2$, $C = 4$, and $C = 8$. The maximum segments length is fixed to $S = 20$.

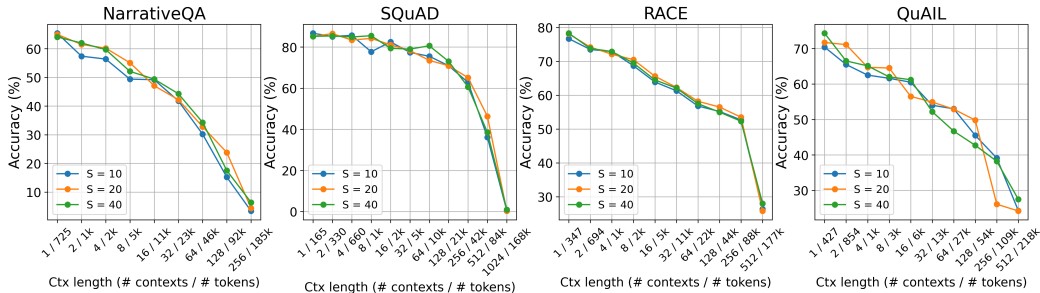

Figure 8: Changing the maximum segments length from $S = 20$ to $S = 10$ and $S = 40$. CompLLM is generally robust to changes of this hyperparameter.

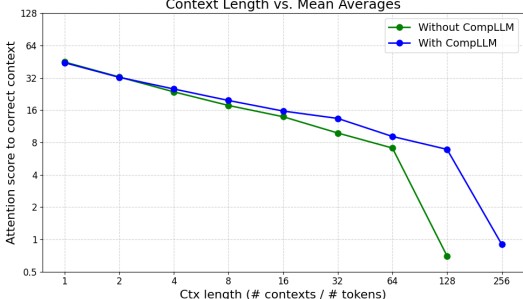

Figure 9: Average attention scores between generated tokens and correct context document on the NarrativeQA dataset. After averaging, we multiply the score by $10^5$ for readability. We can see that the attention score to the correct context document roughly follows the accuracy.

held by Corwin and spot the Creepy Thin Man (Crispin Glover) who was seen on the surveillance videos during Knox's kidnapping. They chase and fight the Creepy Thin Man, but he runs away. When they follow him, they discover Knox. After the Angels reunite Knox with his business partner Vivian Wood (Kelly Lynch), Charlie explains that they must determine whether the Creepy Thin Man has stolen Knox's voice-recognition software. The Angels infiltrate Redstar headquarters, fool the security system, and plant a device in the central computer that will enable them to explore it remotely. They retire for the night after giving Bosley the laptop computer that communicates with the Redstar computer. Dylan takes up Knox's offer to spend the night with him, end up in making love but he betrays her later that night, explaining that he faked the kidnapping with help from Vivian and the Creepy Thin Man. He has kidnapped Bosley, and, with access to Redstar's central computer, he intends to use his voice software with the Redstar satellite network to find and kill Charlie, who he believes had killed his father in the Vietnam War. Knox shoots at Dylan, seemingly killing her, but she escapes unharmed. Natalie and Alex are also attacked, and Corwin is murdered by the Creepy Thin Man. When the Angels regroup together, all uninjured, Charlie's offices are blown up. A radio

receiver survives in the rubble, and Natalie deduces Bosley's location as he speaks to the Angels using a radio transmitter implanted in his teeth, explaining how to spot his location where he is being held captive. With help from Dylan's current boyfriend The Chad (Tom Green), the Angels approach the abandoned lighthouse where Knox is holding Bosley prisoner. The Angels rescue Bosley and defeat Vivian, the Creepy Thin Man, and some henchmen before Knox blows up the lighthouse, but Knox uses his software and the Redstar satellite network to locate Charlie when he telephones Bosley. When Knox programs a helicopter with a missile towards Charlie's house, Bosley helps the Angels board the helicopter, and Alex reprograms the missile to have it shoot backwards, which blows up the helicopter and kills Knox while all of the Angels land safely together on the beach. Seeing the opportunity to finally meet Charlie in person, they enter the beach house that Knox had targeted the missile at, but Charlie has already left. He remotely congratulates the Angels on a job well done through another speaker, and treats them and Bosley to a vacation. Charlie tells them that Knox's father was undercover; however, he was discovered and he was killed by someone else but not Charlie. When he speaks to the Angels unseen again by telephone on the beach, they ask if they could ever meet him in person. Dylan then suspects that she might be seeing him nearby talking into a cell phone, but she doesn't tell the group.

**Question:** How does Charlie communicate with the "Angels"?

**Ground truth answer:** Charlie uses speakers in his offices.

**Generated answer without CompLLM:** Charlie communicates with the "Angels" using a **speaker in his offices**, which allows him to speak directly to them. This speaker is a key element of the story, as it is through this device that Charlie gives instructions, provides updates, and interacts with the Angels throughout the film. Additionally, Charlie is **voiced by John Forsythe** in the film, which means his voice is heard through the speaker, even though he is not physically present on set. The Angels do not see Charlie, but they hear his voice and follow his guidance. So, the answer is: **Charlie communicates with the Angels through a speaker in his office, which is used to speak directly to them.**

**Generated answer with CompLLM:** Charlie communicates with the "Angels" through a **speaker system** in his offices. This speaker allows him to speak directly to the Angels, even though he is not physically present. The communication is done **unseen**, meaning the Angels do not know he is speaking to them unless he chooses to reveal himself. In the film, Charlie is portrayed as an **unseen, omnipresent figure** who controls the Angels from afar. He uses the speaker to give them instructions, relay information, and even offer them a vacation after the mission. His communication is **audio-only**, and he remains hidden from the Angels' view, which adds to the mystery and tension of the story. So, to summarize: - **Method of Communication**: A **speaker system** in Charlie's office. - **Nature of Communication**: **Audio-only**, **unseen**, and **remote**. - **Purpose**: To direct and guide the Angels in their missions.

### A.1.2 EXAMPLE FROM SQUAD

**Context:** Harvard has been highly ranked by many university rankings. In particular, it has consistently topped the Academic Ranking of World Universities (ARWU) since 2003, and the THE World Reputation Rankings since 2011, when the first time such league tables were published. When the QS and Times were published in partnership as the THE-QS World University Rankings during 2004-2009, Harvard had also been regarded the first in every year. The University's undergraduate program has been continuously among the top two in the U.S. News & World Report. In 2014, Harvard topped the University Ranking by Academic Performance (URAP). It was ranked 8th on the 2013-2014 PayScale College Salary Report and 14th on the 2013 PayScale College Education Value Rankings. From a poll done by The Princeton Review, Harvard is the second most commonly named "dream college", both for students and parents in 2013, and was the first nominated by parents in 2009. In 2011, the Mines ParisTech : Professional Ranking World Universities ranked Harvard 1st university in the world in terms of number of alumni holding CEO position in Fortune Global 500 companies.

**Question:** According to the Princeton Review where has Harvard ranked as a "Dream College" in 2013

**Ground truth answer:** Second most commonly

**Generated answer without CompLLM:** According to the Princeton Review, in **2013**, Harvard ranked **second** as a "dream college" among those most commonly named by both students and parents.

**Generated answer with CompLLM:** According to the Princeton Review, in **2013**, **Har-

vard University** was ranked as the **second most commonly named "dream college"** by both students and parents. This means Harvard came in **second** in that particular ranking.

### A.1.3 EXAMPLE FROM RACE

**Context:** The Chinese tradition of giving gifts of money in red envelopes at Lunar New Year has turned into big business for Web giants Alibaba and Tencent, which now both offer electronic "hong bao". At the end of each lunar year, it is common in China to give children hong bao – some money in envelopes that are red, the colour of success. But now theold also take part in this activity. It is now possible to exchange "red envelopes" with smartphone, which is popular in China and has caused a battle for the _ market between the two companies providing the service, Tencent and Alibaba. "You don't have to pay the same cost or wait as long as you would for a traditional bank transfer . It's more convenient, simple and fun," Wang Le, a 28-year-old Beijinger, told AFP. " With electronic red envelopes, you're not limited by your identity or the time of year. It's a new, fresh way of playing the game." The idea was introduced in 2014 by WeChat, a mobile messaging system with over 400 million users and run[:**]by Tencent, China's largest Internet service system. It was successful at once. This year, Alipay Wallet , the payment system run by Alibaba, is ready to take on the competition. It is allowing its 190 million users to send digital gifts, especially on the Twitter-like Sina Weibo. For the past few weeks a digital battle has been terrible, with Tencent banning Alipay from sending red envelopes on WeChat, saying they were at a risk. Recently, WeChat also blocked Alibaba's music app Xiami. To attract the public's attention, the two companies have launched lotteries through which they award red envelopes to users in an online game. It's easy to send and receive hong bao or take part in the lotteries: you simply need to register your bank details. According to market research group iResearch, Alipay controls 82.6 percent of the Chinese mobile phone payment market, compared to 10 percent for Tencent's Tenpay.

**Question:** The following is a multiple choice question (with answers), about the above text. Think step by step and then make sure to end your answer with "the answer is (X)" where X is the correct letter choice. What's the main business purpose for Tencent and Alibaba launch lotteries? Possible answers are:

A: To catch the new fashion on the website.
B: To raise money from netizens.
C: To obtain a better reputation.
D: To gain many more new App users.

**Ground truth answer:** D

**Generated answer without CompLLM:** To determine the main business purpose for Tencent and Alibaba launching lotteries, let's analyze the text step by step: 1. **Context of the Lotteries**: The text mentions that both companies have launched lotteries to "attract the public's attention" and "award red envelopes to users in an online game." This indicates that the lotteries are a marketing or promotional tool. 2. **Purpose of the Lotteries**: The lotteries are described as a way to engage users and increase interaction with their platforms. The text also notes that the companies are competing in the "red envelope market," which is a significant part of their business. 3. **Business Goals**: The primary goal of these lotteries is likely to increase user engagement, encourage the use of their platforms, and gain a competitive edge in the market. This aligns with the idea of attracting more users and keeping them active on the platforms. 4. **Evaluation of Answer Choices**: - **A: To catch the new fashion on the website** – While the lotteries are new and trendy, this is not the main business purpose. - **B: To raise money from netizens** – The lotteries do not directly raise money from users; they are more about engagement than fundraising. - **C: To obtain a better reputation** – This is not mentioned or implied in the text. - **D: To gain many more new App users** – This is the most plausible answer. The lotteries are a way to attract users to their platforms and increase their user base, which is a key business objective. The answer is (D).

**Generated answer with CompLLM:** To determine the main business purpose for Tencent and Alibaba launching lotteries, let's analyze the text step by step: 1. **Context of the Lotteries**: The text mentions that both companies launched lotteries to "attract the public's attention." This is done through online games where users can participate and win red envelopes. 2. **Purpose of the Lotteries**: The lotteries are described as a way to "attract the public's attention" and "to compete in the market." The goal is to engage users and increase their interaction with the platforms. 3. **Business Goals**: The text also mentions that both companies are competing in the "red envelope" market, which is a significant part of the digital payment and social media ecosystem in China. By launching lotteries, they aim to increase user engagement, promote their platforms, and

gain a larger share of the market. 4. **Evaluating the Options**: - **A: To catch the new fashion on the website** – This is somewhat related, but not the main business purpose. - **B: To raise money from netizens** – While lotteries may generate some revenue, the text does not emphasize this as the main purpose. - **C: To obtain a better reputation** – This is not directly mentioned in the text. - **D: To gain many more new App users** – This is the most accurate interpretation. The lotteries are a strategy to attract more users to their platforms, which aligns with their competitive efforts in the red envelope market. Therefore, the main business purpose for Tencent and Alibaba to launch lotteries is to **gain many more new App users**. the answer is (D)

### A.1.4 EXAMPLE FROM QUAIL

**Context:** Seven-hour shifts are really easy to do, especially when you have spent the last year doing only twelve-hour shifts. The jobs tonight were pretty easy - even easier for me as I was driving the ambulance rather than treating the patients. We had a 16-year-old girl with a sore throat, a pair of drunks, one of whom had a twisted ankle, a little old lady who'd fallen over indoors and had a nasty scrape to her arm, and a young woman, twelve weeks pregnant, who had been assaulted at work and struck in the stomach. The real standout job for me shows just how daft some people are. The patient was a twelve-year-old boy. We got the job as 'child banging head on walls and floor' and when we turned up the child was indeed clutching his head and hitting it against a wall. The parents and child spoke poor English, but we easily managed to learn that the child was suffering from an earache, and that this was the cause of the head-hitting. 'How long has he had the pain?' asked my crewmate for the night. 'Five years then, three hours now,' replied the father. We understood what he meant - the child had an earache five years ago, but this current episode, and the reason why we were called out, had lasted three hours. 'Have you given him any painkillers?' 'No,' the father looked confused. 'Do you have any painkillers?' my crewmate asked. 'Yes, but we haven't given him any,' said the father. So the family could see their child rolling around the floor, screaming in pain and banging his head against the walls, and didn't consider that a painkiller might have - oh, I don't know - helped with the pain. I can imagine the scene in the hospital when the nurses give the child some pain relief - the parents looking at each other, slapping their foreheads and saying, 'Doh! We could have done that!' There are a lot of daft people out there - and I get to meet most of them.
**Question:** The following is a multiple choice question (with answers), about the above text. Think step by step and then make sure to end your answer with "the answer is (X)" where X is the correct letter choice. When did the narrator imagine the nurses were giving painkillers to the 12-year old boy
Possible answers are:
A: after talking to the parents,
B: not enough information
C: while talking to the parents
D: before talking to the parents,
**Ground truth answer:** A
**Generated answer without CompLLM:** Let's break this down step by step. The key part of the text is this sentence: ¿ "I can imagine the scene in the hospital when the nurses give the child some pain relief - the parents looking at each other, slapping their foreheads and saying, 'Doh! We could have done that!'" This sentence describes the narrator's imagination of what would happen in the hospital after the nurses give the child painkillers. The parents would realize that they should have given the child painkillers earlier. The question is asking **when** the narrator imagined the nurses giving painkillers to the child. The text says that the parents did not give the child any painkillers, and the narrator is imagining the scenario where the nurses **do** give the child painkillers. This happens **after** the parents have already been talked to, because the parents were the ones who said they didn't give the child any painkillers. So the nurses giving the child painkillers is something the narrator imagines **after** talking to the parents. Therefore, the correct answer is: **A: after talking to the parents** the answer is (A)
**Generated answer with CompLLM:** Let's break down the passage to determine when the narrator imagined the nurses giving painkillers to the 12-year-old boy. The key part of the text is this: ¿ "I can imagine the scene in the hospital when the nurses give the child pain relief, the parents looking at each other, slapping their foreheads and saying, 'Doh. We could have done that!'" This sentence indicates that the narrator is imagining the scenario in the hospital, where the nurses give the child pain relief. The parents, after seeing this, realize they should have given the child painkillers earlier. The question asks: **When did the narrator imagine the nurses were giving painkillers to the 12-

year-old boy?** The passage clearly states that the narrator is imagining the nurses giving the pain relief **in the hospital**, which happens **after** the parents have already been talking to the narrator and the narrator has learned about the child's condition. Therefore, the nurses giving the pain relief is imagined **after** the narrator has talked to the parents. The answer is (A): after talking to the parents. the answer is (A)

