# OpenReview forum: "CompLLM: Compression for Long Context Q&A"
_ICLR.cc/2026/Conference — Submitted to ICLR 2026_

### Official Review · Reviewer_GbqF · 2025-10-20

**Soundness:** 3
**Presentation:** 2
**Contribution:** 2
**Rating:** 4
**Confidence:** 4

**Summary:**

This paper introduce compLLM, a soft compression method to segment and compress extremely long context for Q&A problems. Experiments show significant speedup and memory efficiency, and achieve the same level as results with uncompressed context.

**Strengths:**

1. The research topic is meaningful. The paper is well organized and easy to follow.
2. The formulation of CompLLM is simple but reasonable. Detailed description and explanation in the embedding space reveals the inner mechanism of the proposed method.
3. The author coundct extensive experiments to demonstrate the effectiveness of CompLLM.

**Weaknesses:**

1. The idea of compression is well-studied and tested in many previous works, e.g., [1][2][3], and seems not to be novel.
2. Only QA tasks are evaluated for CompLLM. A significant drawback in compression-based approach is that they may fail in fine-grained retrieval settings, such as Needle-in-haystack (NIAH). Besides, there are many other typical domains that require long-context understanding capabilities, such as code completion and in-context learning. More experimental results should be provided to prove the generalizability of the proposed method.

[1] Chevalier, Alexis, et al. "Adapting language models to compress contexts." arXiv preprint arXiv:2305.14788 (2023).

[2] Han, Wei, et al. "Two are better than one: Context window extension with multi-grained self-injection." arXiv preprint arXiv:2410.19318 (2024).

[3] Zhang, Peitian, et al. "Soaring from 4k to 400k: Extending llm’s context with activation beacon." arXiv preprint arXiv:2401.03462 2.3 (2024): 5.

**Questions:**

See weakness

---

> ### Author Response · Authors · 2025-11-20
> **Response to Reviewer GbqF**
>
> We thank the reviewer for noting that the topic is meaningful and that the extensive experiments demonstrate the effectiveness of CompLLM. While we do agree that context compression for LLMs has been extensively studied, we note that there are still significant open challenges in the field, making research in the field meaningful and with plenty of applications. We should also stress that our goal is not to claim “compression” itself as novel. Unlike previous soft compression techniques, CompLLM’s segment-wise soft compression design and the properties it enables in practical long-context deployments make it unique in this field.
>
> Unlike existing soft-compression methods that compress the entire prompt holistically (either into a short sequence of latent embeddings or into a KV cache), CompLLM (i) splits the context into short segments that are compressed independently, (ii) uses the resulting Concept Embeddings directly as LLM inputs without modifying the LLM weights, and (iii) preserves reusability of compressed segments across queries. This yields linear compression complexity in context length, enables caching of compressed segments, and allows CompLLM to be plugged in or out without changing the base LLM or its training. To the best of our knowledge, no prior soft-compression method combines these three properties—efficiency, scalability, and reusability—in a single framework.
>
> Our work is explicitly scoped to semantic long-context question answering, which we argue are among the most common real-world long-context workloads (e.g., querying large document collections, multi-document reading comprehension). Accordingly, we evaluate CompLLM on four diverse Q&A datasets (NarrativeQA, SQuAD, RACE, QuAIL), training on NarrativeQA+RACE and testing on all four, and on the LOFT long-context benchmark across five datasets, and we do so with two different 4B LLM backbones (Gemma3-4B and Qwen3-4B). In all these settings, a single CompLLM trained once is reused without further tuning, and consistently matches or improves the uncompressed baseline while reducing TTFT by up to 4× and KV cache memory by 2×, demonstrating robustness across models, datasets, and context lengths.
>
> Fine-grained “needle in a haystack” (NIAH) style tasks that require exact token- or character-level retrieval (e.g., counting letters or finding typos) are indeed challenging for every soft compression method, including ours. We explicitly acknowledge this limitation in the paper and emphasize that CompLLM is designed to preserve semantic information rather than exact form, which better reflect its real-world usage. Importantly, because the base LLM is kept frozen and CompLLM is an optional pre-processing module, applications that require exact token-level behavior (NIAH-style probes, strict byte-level retrieval, etc.) can simply bypass compression and run the LLM in standard mode. We will make this connection to NIAH and other fine-grained retrieval probes more explicit in the Limitations and Discussion sections.
>
> Regarding other domains such as code completion and in-context learning, CompLLM is architecturally agnostic to the underlying modality: it operates on token embeddings and can therefore be applied to code or other tokenizations without changes. Our experiments focus on context-based Q&A datasets, in line with virtually every text compression paper. We explicitly acknowledge this in the paper, and leave experiments on other domains to future work.

---

> > ### Author Response · Authors · 2025-11-28
> > **Response**
> >
> > Dear Reviewer,
> >
> > As the discussion period is coming to a close, we wanted to respectfully check if our previous response and the updated manuscript sufficiently addressed your concerns.
> >
> > Thank you

---

> > ### Comment · Reviewer_GbqF · 2025-11-28
> > **Reply to authors**
> >
> > Thanks for the provided information. I think my judgement aligns with the quality of paper. Therefore, I choose to main the score.

---

### Official Review · Reviewer_QjAN · 2025-10-30

**Soundness:** 2
**Presentation:** 2
**Contribution:** 1
**Rating:** 2
**Confidence:** 3

**Summary:**

The paper proposes CompLLM, a soft context compression approach for long-context QA. It segments the input into short chunks and compresses each segment independently into “concept embeddings” (CEs). And then feeding them to an unmodified LLM. The claimed benefits are linear-time compression, scalability to very long contexts, and reusability of compressed segments. With a 2× compression rate, the authors report up to 4×speedups and 2× KV-cache reduction, while maintaining or improving accuracy on long contexts. Experiments cover open-ended and multiple-choice Q&A and a LOFT-style long-context setup.

**Strengths:**

1.	The paper contains a clear complexity analysis (distinguishing KV prefill vs. decoding), and reports TTFT and KV cache savings that match the theoretical scaling

2.	The method description (segmenting, S=20, C=2, LoRA + linear head) is easy to follow.

**Weaknesses:**

1.	Lack of novelty: The core difference from prior soft compression is to compress in segments rather than all long-context. This is only a design hyperparameter (segmenting strategy and segment length S) rather than a fundamentally new modeling principle. A deeper analysis or theory explaining why segment-wise independence preserves task-relevant information—and when it fails—would be needed to elevate the conceptual contribution.

2.	Lack of baselines: The experimental comparison is primarily w/ CompLLM vs. w/o compression. There is no broad comparison with existing compression baselines, which undermines the empirical claims. Strong, relevant baselines such as LLMLingua-2 are absent. Without these, it is unclear whether the reported gains are specific to CompLLM.

3.	“Segment for Long Context” is positioned as the main contribution, but there is no thorough study of how segment length S (and overlap, boundary effects, tokenizer granularity) impacts accuracy, latency, and reuse. An ablation over S, with detailed accuracy/latency/KV trade-offs and error analysis, is missing.

4.	The paper motivates reusability in RAG-style pipelines but does not provide results against standard RAG baselines (e.g., top-k retrieval + reranking + uncompressed prompting). It remains unclear whether CompLLM improves end-to-end RAG outcomes (accuracy, latency, cost) versus optimized retrieval that reduces N in the first place.

**Questions:**

1. Can you add head-to-head comparisons at matched compression rates with strong baselines (e.g., LLMLingua-2)?

2. How does performance vary with segment length (and optional overlap)?

3. Are the method is useful for RAG?

4. How do results scale to larger models and higher compression (C>2)? Is degradation graceful?

---

> ### Author Response · Authors · 2025-11-20
> **Response to Reviewer QjAN**
>
> We should emphasize that design choice of compressing in segments brings fundamental changes to the entire pipeline, when compared to previous works; specifically, it makes CompLLM the only soft compression method that can (1) compress context in linear time, (2) process much longer contexts than those used at train time, and (3) outperform the non-compressed baseline at high context lengths. We strongly believe that its ability to achieve lower latency, lower memory consumption and better results at high context lengths will make CompLLM an essential piece in every future LLM for long-context understanding.
>
> **Other baselines**
>
> A head-to-head comparison with LLMLingua-2 was already present in the main paper in Figure 6 and Section 4.5, making LLMLingua-2 the main competitor of CompLLM. We reported 40 empirical comparisons between LLMLingua-2 and CompLLM, proving that CompLLM largely outperforms LLMLingua-2 in almost every case. Nevertheless, now we also have added comparisons against LongLLMLingua and AutoCompressor, and added them to Figure 6.
>
> **Further ablations**
>
> We have added ablations with varying segment length (Section 4.6 and Figure 8), proving that CompLLM is robust to various values of this hyperparameters (i.e. results are almost unchanged when using S=10, 20, or 40). Similarly, we also dded an ablation with higher compression rates, of 4x and 8x, showing that, as expected, this leads to lower accuracy and larger speedups. We see that even with 8x compression rates CompLLM is competitive with LLMLingua-2 with 2x compression, outperforming it by 4.3% with 1 context on NarrativeQA and underperforming it by 3.7% on SQuAD. We added results and discussion in Section 4.6 and Figure 7.
>
> We should emphasize that LLM compression methods are not replacement to RAG pipelines.
> We therefore compare CompLLM against other compression methods, as well as against the non-compressed baseline, following virtually any previous compression work (Chari et al., 2025; Kim et al., 2024; Liu et al., 2024b; Petrov et al., 2025b; Li et al., 2025; Beltagy et al., 2020; Zaheer et al., 2020)

---

> > ### Author Response · Authors · 2025-11-28
> > **Response**
> >
> > Dear Reviewer,
> >
> > As the discussion period is coming to a close, we wanted to respectfully check if our previous response and the updated manuscript sufficiently addressed your concerns.
> >
> > Thank you

---

### Official Review · Reviewer_1Ki7 · 2025-10-30

**Soundness:** 3
**Presentation:** 3
**Contribution:** 3
**Rating:** 6
**Confidence:** 5

**Summary:**

The authors introduce a technique for compressing transformer context by a factor of 2.  The initial context is first broken into chunks, and each chunk is then compressed separately, using gist tokens.   Compression only requires local attention within the chunk.  The outputs from the gist tokens are then fed in as "soft tokens" to a frozen pre-trained LLM.

The technique reduces prefill costs by (up to) a factor of 4, because attention during prefill is O(N^2) wrt. to sequence length, and the sequence length has been reduced by a factor of 2.  It also reduces inference costs by a factor of 2, since each newly generated token attends over half the sequence length.

On various long-context tasks, compression often /increases/ accuracy.

**Strengths:**

To be honest, I was ready to reject this paper after reading the abstract.  Using gist tokens (i.e. EOS tokens) for in-context compression is a widely used technique -- see the "in-context autoencoder," and "getting to the gist of gisting" (cited by the authors).  Dividing the input into segments is also hardly new; sliding window attention is widely used, and other works (e.g. Melodi), not only do compression at the segment level, but use gist tokens to implement the compression.

However, the authors have made a number of well-thought-out design decisions in this paper, and the total is more than the sum of its parts.  The main contributions as I see them are:

* Uses a frozen pre-trained LLM as-is, which is very important.
* The compression model is the *same* pre-trained LLM, fine-tuned with LORA so that they share weights.
* Fine-tuning splits the prompt into context/question/answer, and the distillation loss measures changes in the *hidden states* of all layers for the answer, rather than just the final output.
* The authors normalize hidden activations, and use a smooth-L1 loss.

These are all non-obvious details that are crucial to the success of the work, IMO.

IMO, the reason for the *increase* in accuracy is probably because the main LLM is run twice -- once to compress, and once to process the context, and thus has twice the depth.  That's a clever way to reduce any negative impacts from compression.

**Weaknesses:**

The authors' claims of "4x reduction in prefill costs" are theoretical, and over-stated.   At small context lengths, prefill is actually 2x *more* expensive -- there's a 50% increase from the gist tokens, and another 50% at having to run both the compression and the frozen LLM.  This should be discussed.  The O(N) parts of an LLM are actually quite expensive -- the feed-forward-blocks are large, and in a model like gemini, many layers use local attention, so global attention dominates only at very long sequence lengths.

Moreover, most uses cases for very long contexts involve many documents concatenated together, in which case block-sparse attention can be used -- a local variant which only attends within each document.  This needs to be discussed.

Given that prefill is 2x more expensive at small context lengths, and potentially 4x less expensive at large lengths, I am most interested in the crossover point -- the point where compression actually starts providing an advantage.  Unfortunately, the authors' charts are scaled in such a way that you can't see this point.

The overall structure of CompLLM is actually quite similar to a number of earlier works on sparse attention patterns; what is essentially happening is two-level attention -- global attention attends only to gist tokens, which in turn attend locally to the original context.  That should be in the discussion and lit review.

**Questions:**

What is the crossover point in sequence length?

A 2x compression rate is not very high -- I would like to see some ablations of what happens with a 4x or 8x rate.

The "getting to the gist of gisting" paper found benefits from evenly spacing the gist tokens throughout the sequence -- it would be nice to see that as an ablation.

---

> ### Author Response · Authors · 2025-11-20
> **Response to Reviewer 1Ki7**
>
> We thank the reviewer for taking the time to thoroughly read the paper, and for sharing some interesting suggestions: we have applied the suggested changes, and believe it now leads to a stronger paper.
>
> **Crossover point**
>
> To better understand under which scenarios is CompLLM providing speedups, we have added a plot (Figure 4, right) showing the crossover point: with Gemma3-4B and default hyperparameters (S=20 and C = 2) it lies at 8k context length, implying that for shorter context lengths one should use CompLLM only if the context can be compressed offline. We emphasize that at longer context lengths (e.g. higher than 16k) CompLLM leads to lower latency, lower memory usage, and better results, suggesting that it will become a core component of any LLM that needs to process long contexts. Crucially, the condition to check if CompLLM should or should not be used for a specific query is as simple as a single line of code (if context length longer than a threshold), making it of trivial implementation in practice.
> We also explicitly mention that (online) CompLLM slows down inference for very short contexts at the end of Section 3.3.
>
> **Compression rates**
>
> We added an ablation with higher compression rates, of 4x and 8x, showing that, as expected, this leads to lower accuracy and larger speedups. We see that even with 8x compression rates CompLLM is competitive with LLMLingua-2 with 2x compression, outperforming it by 4.3% with 1 context on NarrativeQA and underperforming it by 3.7% on SQuAD. We added results and discussion in Section 4.6 and Figure 7.
>
> **Interleaved embeddings**
>
> As suggested by the reviewer, we ran ablation by interleaving the concept embeddings throughout the segment. We found it to underperform the default version of CompLLM by 3.8% on average across the 4 test datasets across all context lengths. A reason for this result is that the first concept embedding can only compress (or attend to) the first few tokens, lacking the ability to see the whole context of the segment; similarly the last tokens can only be compressed by the last concept embedding, which might be a complex task in case those last few tokens are highly informational and cannot be compressed in just a single embedding. We also note that in Gist of Gisting the authors compress the whole context as a whole: in this case if all gist tokens are added at the end, they might be far from the initial tokens and might have a hard time attending to them. This does not happen in CompLLM, as the compression is performed in small chunks.
>
> **Sparse attention**
>
> The reviewer raises an insightful parallel between CompLLM and sparse attention mechanisms, specifically regarding the use of block-sparse attention for concatenated documents, and we have updated Section 2.2 to discuss these connections. While computational complexity reduction is a shared benefit between CompLLM and sparse attention methods, CompLLM offers distinct practical advantages over standard sparse attention approaches, which often rely on dynamic pattern estimation during inference and do not inherently compress the memory footprint of the KV cache. In contrast, CompLLM physically reduces the sequence length, guaranteeing a reduction in KV cache size, such as a 50% reduction with a 2x compression rate, and utilizes a query-independent compression scheme that allows segments to be processed offline and cached for reuse across multiple queries , thereby avoiding the online overhead associated with calculating dynamic sparse indices for every new input.

---

> > ### Comment · Reviewer_1Ki7 · 2025-11-21
> >
> > Thank you for the update.  An 8k crossover point is pretty much what I would expect, and I'm glad to see the compression rate ablations.  :-)  I am raising my score.

---

### Official Review · Reviewer_v9hg · 2025-10-31

**Soundness:** 2
**Presentation:** 3
**Contribution:** 2
**Rating:** 4
**Confidence:** 3

**Summary:**

The paper proposes CompLLM, a framework for efficient long-context question answering by segment-wise soft compression. Long documents are divided into segments; each segment is independently compressed into a small set of concept embeddings (CEs) using a lightweight compressor model. The CEs, together with the user query, are fed to a frozen LLM to generate answers. This segmentation enables linear-time compression, parallel processing, and caching of compressed segments for reuse across queries. Empirical results show comparable or improved accuracy to uncompressed baselines, with substantial reductions in KV-cache size and latency. While the system demonstrates strong practical gains, the conceptual novelty beyond prior work such as AutoCompressor and LLMLingua is limited.

**Strengths:**

- Simple, scalable design that makes soft-compression linear-time and cacheable.
- Strong empirical performance on long-context QA benchmarks with clear latency and memory gains.
- Compatible with frozen LLMs, making the method easy to integrate into real systems.

**Weaknesses:**

- Conceptual novelty is limited — largely a segment-wise parallel variant of AutoCompressor.

- Baseline coverage is only partial: while the paper includes LLMLingua2 as a baseline, it omits comparisons to other recent or stronger soft-compression and pruning methods (e.g., AutoCompressor, LongLLMLingua, or token-selection approaches).

- Lacks deeper analysis or theory explaining why compression sometimes improves accuracy (e.g., “attention dilution” remains a hypothesis).

**Questions:**

How does CompLLM perform on tasks requiring fine-grained token fidelity (e.g., code, math, counting)? Including such cases would better delineate the method’s limitations.

---

> ### Author Response · Authors · 2025-11-20
> **Response to Reviewer v9hg**
>
> We thank the reviewer for acknowledging the scalable design and strong performance; we took note of the reported weaknesses, and used the advice to improve the current version of the paper, which addresses the reviewer suggestions.
>
> **Baselines and empirical coverage**
>
> We agree that the original baseline set was incomplete. We now include AutoCompressor and LongLLMLingua at matched compression ratios. Both underperform CompLLM on most long-context QA benchmarks at similar compression levels, despite AutoCompressor running multiple iterative passes over the context, and LongLLMLingua being question-aware (i.e. if a follow-up question is asked, LongLLMLingua compression needs to be run again). These results reinforce our main claim that CompLLM matches or improves uncompressed accuracy while delivering strong KV-cache and latency savings, and is competitive or superior to existing soft-compression and pruning approaches.
>
> **Differences from AutoCompressor and LLMLingua-style methods**
>
> We respectfully disagree that CompLLM is “largely a segment-wise parallel variant of AutoCompressor,” and have clarified the distinctions, also updating Section 2.2:
>
> * CompLLM trains a separate lightweight compressor via distillation from a frozen LLM; AutoCompressor updates the LLM itself.
> * CompLLM uses a distillation paradigm and applies a SmoothL1 loss on the activations of self-generated tokens; AutoCompressor uses a cross-entropy loss on ground-truth tokens.
> * CompLLM keeps LLM weights frozen and is plug-and-play with off-the-shelf models; AutoCompressor requires LLM fine-tuning.
> * CompLLM produces segment-local concept embeddings that are cacheable and reusable across arbitrary future queries; AutoCompressor’s compressed tokens depend on all preceding tokens, preventing simple reuse.
> * CompLLM requires only two forward passes per token (compressor + LLM) and compresses segments fully in parallel; AutoCompressor uses multiple sequential passes over a growing prefix.
>
> **Analysis of attention dilution**
>
> The revised version adds an analysis of how attention on relevant context spans varies with context length, with and without CompLLM (Section 4.6 and Figure 9). We observe that uncompressed attention becomes more diffuse as context grows, while CompLLM concentrates attention on CEs that summarize relevant content. This provides empirical support to our claim that CompLLM reduces attention dilution.
>
> **Scope and fine-grained fidelity tasks**
>
> Following a large number of papers on context compression, such as the aforementioned AutoCompressor, LLMLingua-2 and LongLLMLingua, our current training data and benchmarks focus on natural-language long-context QA, although CompLLM (as well as previous compression methods) is architecturally agnostic to the underlying domain.
> It is important to note that, just like any other compression method and LLMs in general, CompLLM cannot handle unseen tokens: for example, being trained only on English language, it would not be able to compress Chinese characters or coding. We have now added this insight in the Limitations section of the paper, and we emphasize that CompLLM can simply overcome this issue by not using the compression in out-of-distribution cases (e.g. tokens unseen during training), whereas many existing soft compression methods can’t skip the compression stage as they do not preserve the original LLM’s weights.

---

> > ### Author Response · Authors · 2025-11-28
> > **Follow up**
> >
> > Dear Reviewer,
> >
> > As the discussion period is coming to a close, we wanted to respectfully check if our previous response and the updated manuscript sufficiently addressed your concerns.
> >
> > Thank you

---

### Meta-Review · Area_Chair_ewfd · 2026-01-07

**Summary:**

This paper presents CompLLM, a segment-wise soft compression method for long-context question answering that achieves linear compression complexity, enables caching of compressed segments, and works with frozen LLMs. The reviewers acknowledged the practical benefits including reduced latency and KV cache size, and the strong empirical results. However, concerns were raised regarding the degree of conceptual novelty and the experimental scope being limited to QA tasks. The authors provided substantial revisions during rebuttal, adding baselines (AutoCompressor, LongLLMLingua), ablations on compression rates and segment lengths, and attention dilution analysis. Unfortunately, the discussion was hampered by two reviewers (v9hg and QjAN) failing to engage with the rebuttal, leaving uncertainty about whether their concerns were resolved. While the practical contributions are notable, the combination of unresolved novelty concerns and incomplete reviewer engagement makes it difficult to recommend acceptance.

**Reviewer Concerns:**

Reviewer 1Ki7's concerns were fully addressed through the added crossover point analysis, compression rate ablations, and clarifications on sparse attention connections. For Reviewer v9hg, the authors added the requested baselines and attention dilution analysis, and provided detailed arguments distinguishing CompLLM from AutoCompressor (different training paradigm, loss function, frozen LLM compatibility, cacheability, and parallelism); however, the reviewer did not respond to evaluate these arguments. Reviewer QjAN's request for LLMLingua-2 comparison was based on overlooking existing results in the paper, and their other concerns about ablations were addressed, but they also did not engage. Reviewer GbqF maintained their score after rebuttal, citing concerns about novelty and generalizability. The outstanding issue is whether the segment-wise design with its practical benefits constitutes sufficient contribution given prior work on soft compression, a question that remains contested.

**Reviewer Scores:**

Reviewer 1Ki7 (Rating 6): Would likely maintain 6 or increase slightly, as their technical concerns were thoroughly addressed.

Reviewer v9hg (Rating 4): Uncertain due to non-engagement. The authors provided substantive responses to the novelty concern with detailed technical distinctions from AutoCompressor, which could have moved the score to 5.

Reviewer QjAN (Rating 2): Uncertain due to non-engagement. Given the poor review quality noted by authors (requesting comparisons already present), this score may not accurately reflect the paper's merit.

Reviewer GbqF (Rating 4): Confirmed to remain at 4 after seeing the rebuttal.

---

### Decision · Program_Chairs · 2026-01-26

Reject